# Direct SARS-CoV-2 infection of the human inner ear may underlie COVID-19-associated audiovestibular dysfunction

Minjin Jeong[1,2,12], Karen E. Ocwieja[3,4,5,13], Dongjun Han[1,2,12,13], P. Ashley Wackym [6], Yichen Zhang [7], Alyssa Brown [2], Cynthia Moncada[2], Andrea Vambutas [8], Theodore Kanne[9], Rachel Crain[10], Noah Siegel[1,2], Valerie Leger[5], Felipe Santos[1,2], D. Bradley Welling [1,2], Lee Gehrke[5,7,11,14 ✉] & Konstantina M. Stankovic[1,2,12,14 ✉]

## Abstract

**Background** COVID-19 is a pandemic respiratory and vascular disease caused by SARS-CoV-2 virus. There is a growing number of sensory deficits associated with COVID-19 and molecular mechanisms underlying these deficits are incompletely understood.

**Methods** We report a series of ten COVID-19 patients with audiovestibular symptoms such as hearing loss, vestibular dysfunction and tinnitus. To investigate the causal relationship between SARS-CoV-2 and audiovestibular dysfunction, we examine human inner ear tissue, human inner ear in vitro cellular models, and mouse inner ear tissue.

**Results** We demonstrate that adult human inner ear tissue co-expresses the angiotensin-converting enzyme 2 (ACE2) receptor for SARS-CoV-2 virus, and the transmembrane protease serine 2 (TMPRSS2) and FURIN cofactors required for virus entry. Furthermore, hair cells and Schwann cells in explanted human vestibular tissue can be infected by SARS-CoV-2, as demonstrated by confocal microscopy. We establish three human induced pluripotent stem cell (hiPSC)-derived in vitro models of the inner ear for infection: two-dimensional otic prosensory cells (OPCs) and Schwann cell precursors (SCPs), and three-dimensional inner ear organoids. Both OPCs and SCPs express ACE2, TMPRSS2, and FURIN, with lower ACE2 and FURIN expression in SCPs. OPCs are permissive to SARS-CoV-2 infection; lower infection rates exist in isogenic SCPs. The inner ear organoids show that hair cells express ACE2 and are targets for SARS-CoV-2.

**Conclusions** Our results provide mechanistic explanations of audiovestibular dysfunction in COVID-19 patients and introduce hiPSC-derived systems for studying infectious human otologic disease.

**Plain language summary**

Coronavirus disease 2019 (COVID-19) is an infectious disease caused by the novel coronavirus SARS-CoV-2. A growing number of sensory symptoms have been linked to this illness. Here, we describe patients with COVID-19 and new-onset of hearing loss, tinnitus and/or dizziness. To examine the underlying molecular mechanisms of these symptoms, we studied human and mouse inner ear tissue. We also generated some of the first human cellular models of infectious inner ear disease. We show that human and mouse inner ear cells have the molecular machinery to allow SARS-CoV-2 entry. We further show that SARS-CoV-2 can infect specific human inner ear cell types. Our findings suggest that inner ear infection may underlie COVID-19-associated problems with hearing and balance.

[1] Department of Otolaryngology—Head and Neck Surgery, Harvard Medical School, Boston, MA, USA. [2] Eaton Peabody Laboratories and Department of Otolaryngology—Head and Neck Surgery, Massachusetts Eye and Ear, Boston, MA, USA. [3] Department of Pediatrics, Harvard Medical School, Boston, MA, USA. [4] Department of Pediatrics, Boston Children's Hospital, Boston, MA, USA. [5] Institute for Medical Engineering and Science, Massachusetts Institute of Technology, Boston, MA, USA. [6] Department of Otolaryngology—Head and Neck Surgery, Rutgers Robert Wood Johnson Medical School, New Brunswick, NJ, USA. [7] Harvard-MIT Program in Health Science and Technology, Boston, MA, USA. [8] Department of Otolaryngology Head and Neck Surgery, Donald and Barbara Zucker School of Medicine at Hofstra/Northwell Health, New Hyde Park, NY, USA. [9] ENT and Allergy Associates of South Georgia, Valdosta, GA, USA. [10] Brevard ENT Center, Rockledge, FL, USA. [11] Department of Microbiology, Harvard Medical School, Boston, MA, USA. [12] Present address: Department of Otolaryngology—Head and Neck Surgery, Stanford University School of Medicine, Stanford, CA, USA. [13] These authors contributed equally: Karen E. Ocwieja, Dongjun Han. [14] These authors jointly supervised this work: Lee Gehrke, Konstantina M. Stankovic. ✉email: lgehrke@mit.edu; kstankovic@stanford.edu

Viral infections are a common reason for hearing loss and vestibular dysfunction. Viruses known to cause audio-vestibular dysfunction include members of the *Herpes-viridae* (cytomegalovirus, herpes simplex virus, varicella zoster virus, Epstein-Barr virus), *Paramyxoviridae* (parainfluenza viruses, mumps virus, measles virus), polio virus, hepatitis viruses, human immunodeficiency virus, rubella virus, and influenza viruses[1]. Presumed mechanisms of virally induced sensorineural hearing loss (SNHL), tinnitus, vertigo, dizziness, or imbalance include: direct invasion and damage of inner ear structures, including the organ of Corti (OC) and the vestibulocochlear nerve[2]; immune-mediated damage and inflammation, including neuroinflammation[3,4]; and reactivation of latent virus within the inner ear[5]. In addition, viruses can infect the middle ear and typically cause conductive hearing loss due to a middle ear effusion. While coronaviruses are a common cause of middle ear infection[6], their role in inner ear infection has not been systematically studied.

Coronavirus disease 2019 (COVID-19) is a contagious respiratory and vascular disease caused by SARS-CoV-2. The virus is also known to cause anosmia[7] and ageusia[8], highlighting its tropism for sensory systems. Although there are several recent reports of audiovestibular symptoms in COVID-19 patients, these rely on self-reported hearing loss[9], do not comment on hearing outcome after COVID-19 resolution[10–13], lack documentation of objective SARS-CoV-2 testing[10], or only include single patients[10,12–17]. Here we provide the largest series to date of patients with documented SNHL and audiovestibular symptoms during SARS-CoV-2 infection, along with audiograms at 2–4 months after the resolution of COVID-19. To investigate whether these symptoms might be due to direct infection of audiovestibular structures, we examined the expression of SARS-CoV-2 cell entry-related genes and proteins in human and mouse inner ear tissue, and we infected human vestibular tissue to identify target cell types of SARS-CoV-2. Finally, in order to study the pathogenesis of COVID-19 audiovestibular symptoms, we derived otic prosensory cells (OPCs) (the precursors of hair cells and their supporting cells), Schwann cell precursors (SCPs) (the progenitor of Schwann cells), and inner ear organoids (three-dimensional (3D) organized structures mimicking inner ear epithelium) from human induced pluripotent stem cells (hiPSCs) and demonstrated differential expression of host-derived SARS-CoV-2 entry cofactors that might influence tropism of the virus in the inner ear.

## Methods

**Human subjects**. The study of patients with COVID-19-associated sudden audiovestibular symptoms was approved by the Massachusetts General Brigham Institutional Review Board (2020P002900). Verbal informed consent was obtained from all subjects and all procedures were conducted in accordance with the Helsinki Declaration. Verbal informed consent was obtained to publish the detailed case information contained within the Supplementary Information. To evaluate the expression of SARS-CoV-2 receptors in the human inner ear, fresh inner ear tissue was collected during surgical labyrinthectomies and translabyrinthine resections of vestibular schwannomas ($N = 6$). This research was reviewed by the Massachusetts General Brigham Institutional Review Board (IRB) and determined to be Exempt from IRB approval (2020P003329) and from the need for informed consent.

**Generation of hiPSC**. For the generation of hiPSC line SK8-A, we recruited healthy subjects and isolated primary dermal fibroblasts from them. The protocols for research involving human subjects and for stem cell research were approved by the

Institutional Review Board of Massachusetts Eye and Ear and Partners Human Research Committee. The study participants provided written informed consent.

For establishing fibroblast lines from skin biopsies, tissue was manually dissected into pieces of approximately 1 cm$^2$ in size and digested with Trypsin (Sigma) for 20 min at room temperature. Digested tissue was centrifuged and incubated with 0.04 mg/ml DNase (Sigma) for 10 min at room temperature. Tissue was collected by centrifugation and further disaggregated by incubation with 20 mg/ml collagenase type II (Gibco) for 30 min at room temperature. Cells were plated onto plates in fibroblast medium containing DMEM (Gibco, Cat# 11995073) supplemented with 10% fetal bovine serum (FBS; Gibco, Cat# 26140079) and 2 mM L-glutamine (Gibco, Cat# 25030081).

One day before transduction, 150,000 mycoplasma-free fibroblasts were seeded per well in a 6-well plate previously coated with 0.1% gelatin and cultured in fibroblast medium. On the day of transduction, cells were transduced with the CytoTune™-iPS 2.0 Sendai Reprogramming Kit (Invitrogen, Cat# A16517) in fibroblast medium at a multiplicity of infection (MOI) of 3 for Klf4 and 5 for Klf4–Oct3/4–Sox2 and cMyc. The medium was replaced with fresh fibroblast medium every day. Five days after transduction, 250,000 cells were transferred onto irradiated mouse embryonic fibroblasts (MEFs) in a 10 cm culture dishes. After culturing overnight in fibroblast media, the medium was replaced daily with DMEM/F12 (Gibco, Cat# 11330057) supplemented with 20% KnockOut Serum Replacement (Gibco, Cat# 10828028), 2 mM L-glutamine, 1× MEM Non-Essential Amino Acids Solution (Gibco, Cat# 11140050), 55 μM β-Mercaptoethanol (Gibco, Cat# 21985023), and 10 ng/ml FGF-2 (Gibco, Cat# PHG0360). The hiPSC colonies were picked for expansion and characterization from 18 to 25 days after transduction.

**Karyotyping/mycoplasma test**. Karyotype analyses were performed at WiCell, according to the International System for Human Cytogenetic Nomenclature. The cell lines were tested at passage 5, with 20 cells in metaphase counted for the analysis. Mycoplasma test was performed using the MycoAlert Mycoplasma Detection Kit (Lonza, Cat# LT07-318) to ensure that all cells are mycoplasma free.

**Embryoid body (EB) formation**. hiPSC were differentiated as EBs by detaching ~80% confluent hiPSC colonies from MEF feeders using Gentle Cell Dissociation Reagent (StemCell Technologies, Cat# 07174) and a cell scraper. Gently detached cells were transferred to a 15 ml centrifuge tube and the cell clumps were allowed to sink for 10–15 min. EBs were transferred to ultra-low attachment 6-well plates (Corning, Cat# 3471) in DMEM/F12 supplemented with 10% KnockOut Serum Replacement and 10 μM Y-27632 (Calbiochem, Cat# 688001); the medium was replaced every other day. Eight days after plating, EBs were transferred to a gelatin-coated 10 cm cell culture dish in DMEM supplemented with 10% FBS and 2 mM L-glutamine. After 7 days of culturing, EBs were harvested for further analysis.

**hiPSCs culture**. hiPSC lines SK8-A and UCSD112i-2-11 (WiCell) on Matrigel hESC qualified matrix (Corning, Cat# 354277) were cultured in mTeSR Plus medium (StemCell Technologies, Cat# 100-0276). hiPSC colonies were treated with ReLeSR (StemCell Technologies, Cat# 05872) and detached by tapping the side of the plates. Detached cell clumps were plated on the Matrigel-coated plate. hiPSC were maintained under mTeSR Plus medium and used before passage number 50.

**OPC differentiation**. We modified the previously published protocol for differentiation of hiPSCs into OPCs using a mono-layer culture system[18]. In our study, undifferentiated hiPSC lines UCSD112i-2-11 and SK8-A were dissociated with ReLeSR and seeded at 30,000 cells/cm$^2$ onto laminin-coated plates (R&D Systems, Cat# 3401-010-02) and cultured in DMEM/F12 (Gibco, Cat# 11330032) supplemented with 1× N2 (1% (v/v) final concentration, Gibco, Cat# 17502048), 1× B27 (2% (v/v) final concentration, Gibco, Cat# 17504044), 50 ng/ml FGF-3 (R&D Systems, Cat# 1206-F3), and 50 ng/ml FGF-10 (R&D Systems, Cat# 345-FG). The medium was replaced on day 1 and changed every other day. The concentration of Y-27632 (TOCRIS, Cat# 1254) was maintained at 10 μM throughout days 0–3. On day 14, the cells were transferred onto growth factor reduced (GFR) Matrigel (Corning, Cat# 356230)-coated plates at 80,000 cells/cm$^2$ and cultured in DMEM/F12 supplemented with 1× N2, 1× B27, 5 μM Dibenzazepine (DBZ; TOCRIS, Cat# 4489), and 10 μM Y-27632. The medium without Y-27632 was replaced every other day from day 15 to day 20.

**SCP differentiation**. For SCP differentiation, we modified a previously published protocol[19]. hiPSC line SK8-A (~10,000 cells/cm$^2$) was replated onto GFR Matrigel-coated culture dishes with 10 μM of Y-27632. The next day, the culture medium was switched from hiPSC culture medium to 10 μM Y-27632 and 10 μM valproic acid (VPA; Sigma, Cat #P4543) supplemented Neuronal Differentiation Medium (NDM). Supplemented NDM contained 1× N2, 1× B27, 0.005% bovine serum albumin (BSA; Sigma, Cat# A8412), 2 mM GlutaMAX (Gibco, Cat# 35050061), 0.11 mM β-mercaptoethanol, 3 μM CHIR99021 (TOCRIS, Cat# 4423), 20 μM SB-431542 (Selleckchem, Cat# S1067) in Advanced DMEM/F12 (Gibco, Cat# 12634028), and Neurobasal medium (Gibco, Cat# 21103049) (1:1 mix). After 2 days of differentiation, the medium was replaced with NDM and freshly changed every other day. After ~2 weeks of differentiation, the cells were dissociated with Accutase (StemCell Technologies, Cat# 07920) and cultured in Neuregulin-1 (NRG-1; R&D Systems, Cat# 5898-NR-050) containing NDM (=SCPM). The cells were split and replated every 3~4 days by using Accutase. The hiPSC-derived SCPs were generated after approximately 20~30 days of differentiation.

**Inner ear organoid differentiation**. We modified the previously published protocol for the generation of inner ear organoids containing vestibular hair cell-like cells from human pluripotent stem cells[20–22]. In our study, hiPSCs from SK8-A line were dissociated with ReLeSR and distributed 5,000 cells per well onto low-adhesion 96-well U-bottom plates in mTeSR Plus medium containing 20 μM Y-27632 and 100 μg/ml Normocin (Invivogen, Cat# ant-nr-2). The plate was centrifuged at 300 *g* for 3 min to aggregate the cells. On day 2, the medium was changed into 100 μl of chemically defined medium (CDM) containing 4 ng/ml FGF-2 (R&D Systems, Cat# 233-FB), 10 μM SB-431542 (TOCRIS, Cat# 1614), 2.5 ng/ml BMP4 (R&D Systems, Cat# 314-BP), and 2% GFR Matrigel. CDM contained a 50:50 mixture of F-12 Nutrient Mixture with GlutaMAX (Gibco) and Iscove's Modified Dulbecco's Medium with GlutaMAX (IMDM; Gibco, Cat# 31980030) additionally supplemented with 0.5% BSA, 1× Chemically Defined Lipid Concentrate (Invitrogen, Cat# 11905031), 7 μg/ml Insulin (Sigma, Cat# I9278), 15 μg/ml Transferrin (Sigma, Cat# T8158), 450 μM Mono-Thioglycerol (Sigma, Cat# M6145), and Normocin. On day 6, 25 μl of CDM containing a 250 ng/ml FGF-2 (50 ng/ml final concentration) and 1 μM LDN-193189 (200 nM final concentration; Stemgent, Cat# 04-0074-02) was added to the preexisting 100 μl of media in each well. On day 10, 25 μl of CDM containing a 18 μM CHIR99021

(3 μM final concentration; Stemgent, 04-0004-02) was added to the preexisting 125 μl of media. On day 14, the aggregates were pooled together and washed with freshly prepared Organoid Maturation Medium (OMM) containing a 50:50 mixture of Advanced DMEM/F12 and Neurobasal medium supplemented with 0.5× N2 Supplement, 0.5× B27 without Vitamin A (Gibco, Cat# 12587010), 1× GlutaMAX, 0.1 mM β-Mercaptoethanol (Gibco), and Normocin. The aggregates were plated individually into each well of a 24-well low-cell-adhesion plate in OMM containing 3 μM CHIR99021 and 1% GFR Matrigel. On day 17, the medium was changed completely with OMM containing 3 μM CHIR99021. On day 20, the medium was changed with OMM. The 24-well plates were maintained on an in-incubator stir plate at 65 RPM for up to 90 days.

**Virus and infections**. The SARS-CoV-2 strain, WA1/2020, was obtained from BEI resources (NR-52281) and propagated in Vero E6 (ATCC) cells in DMEM (Corning) with 2% FBS (Gibco) with penicillin and streptomycin (Gibco, Cat# 15140122). Virus-containing cell supernatants were harvested after the appearance of cytopathic effect and clarified by centrifugation. Conditioned media was collected from uninfected Vero E6 cells grown in parallel. Viral titer was determined by plaque-forming assay in Vero E6 cells: cells were infected with dilutions of viral stock and overlaid with 3.2% carboxymethylcellulose solution mixed at 1:1 with DMEM containing 4% FBS. At 4 days post infection, cells were fixed in methanol and stained with 0.5% crystal violet.

Adult human vestibular tissue was maintained in explant media: Neurobasal Medium supplemented with 1× N2, 2× B27, 1 mM GlutaMAX, 0.5 mM dibutyryl-cyclic AMP (Santa Cruz Biotechnology, Cat# sc-201567B), 10 ng/ml human brain-derived neurotrophic factor (R&D Systems, Cat# 248-BDB), 10 ng/ml human neurotrophin-3 (R&D Systems, Cat# 267-N3), 10 ng/ml insulin-like growth factor-1 (R&D Systems, Cat# 291-G1), and 50 μg/ml Normocin. For infection of explants, at 1 day post tissue harvest, media was exchanged for SARS-CoV-2 inoculum at a viral concentration of 5e+04 pfu/ml (a total of 2.5e+04 pfu per explant). Mock infections were performed with an equivalent dilution of conditioned media. The tissues were incubated for 2 h at 37°C with manual rocking after which viral inoculum was removed and replaced with fresh explant media. At 48 h after media change (48 hours post infection, hpi), the tissues were fixed in 4% paraformaldehyde (PFA) for 4 h. The tissues were washed three times in phosphate-buffered saline (PBS) and then dehydrated overnight in 10% sucrose followed by in 20% sucrose and 30% sucrose. The tissues were embedded in OCT compound (Fisher HealthCare, Cat# 23-730-571) and frozen for sectioning.

For 2D infections, OPCs and SCPs were seeded in black-walled 96-well plates (Corning, Cat# 3603) and infected at day 23 and 21–31 post differentiation, respectively. Virus was added to cells in minimal volume at indicated MOIs diluted in DMEM/F12 (mock controls received equal volume of conditioned media) and incubated with intermittent rocking for 1 h at 37°C. Inoculum was then removed and replaced with fresh OPC/SCP medium. For immunofluorescence studies, at the indicated time post infection, cells were fixed in 3.2% PFA (Electron Microscopy Sciences, Cat# 15714) in PBS for 30 min at room temperature. After fixation, cells were washed three times with PBS. All work with infectious materials was performed in the BSL3 facility at the Ragon Institute in accordance with approved protocols at that facility.

For 3D infections, organoid media was partially exchanged for diluted SARS-CoV-2 at a final viral concentration of 5e+04 pfu/ml (a total of 2.5e+04 pfu per organoid) in fresh organoid media. Mock infections were performed with an equivalent dilution of

conditioned media. Organoids were incubated with inoculum overnight at 37°C with orbital shaking (65 rpm), after which the organoids were moved to fresh media. At 48 h after media change (72 hpi), organoids were fixed in 4% PFA for 45 min. Organoids were washed three times in PBS and then dehydrated overnight in 15% sucrose followed by 5 days in 30% sucrose. Organoids were embedded in OCT compound and frozen in 2-methylbutane for sectioning.

**Viability assays**. The CellTiter-Glo® Luminescent Cell Viability Assay (Promega, Cat# G7570) was used according to the package protocol. Cells were lysed in situ in the 96-well plate in which they were grown, and then the material was transferred to a 96-Well Solid White Polystyrene Microplate (Corning, Cat # 3912) for luminescence reading.

**Mouse inner ear collection**. Wild-type CBA/CaJ mice were obtained from the Jackson Laboratory (Bar Harbor, ME, USA). Animals of either sex were used for experimentation in an estimated 50:50 ratio. Six weeks old mice were sacrificed through cardiac perfusion, decapitated, and inner ears were removed. Inner ears were then fixed in buffered 4% PFA (ThermoFisher Scientific, Cat# AAJ19943K2) for 2–3 h after piercing both the round and oval windows, decalcified in 0.12 M EDTA at room temperature for 4 days, serially dehydrated in sucrose, embedded in OCT compound, and sectioned into 12 μm cryosections on a Leica CM-1860 cryostat. This research was approved by the Institutional Animal Care and Use Committees at Massachusetts Eye and Ear.

**qRT-PCR**. To isolate RNA from human vestibular specimens, the tissue was disrupted and homogenized using the TissueRuptor and RNeasy Plus Micro Kit (QIAGEN, Cat# 74034). RNA from the cell cultures in vitro was isolated using the ReliaPrep™ RNA Cell Miniprep System (Promega, Cat# Z6011). Single-stranded complementary DNA was synthesized from 1 μg of the RNA sample, or less than 1 μg of RNA in case of human surgical specimens, using GoScript Reverse Transcription System (Promega, Cat# A5000). The quantitative reverse transcription PCR (qRT-PCR) was performed with the GoTaq qPCR Master Mix (Promega, Cat# A6002) in a QuantStudio 3 real-time PCR system (Applied Biosystems). The reaction parameters were as follows: 95°C for 2 min to denature the cDNA and primers, 40 cycles at 95°C for 15 s, annealing/extension at 60°C for 60 s. A comparative Ct method was used to calculate the levels of relative expression, whereby the Ct was normalized to the endogenous control glyceraldehyde-3-phosphate dehydrogenase (GAPDH). This calculation gave the ΔCt value, which was then normalized to a reference sample (i.e., a negative control), giving the ΔΔCt. The fold change was calculated using the following formula: $2^{-\Delta\Delta Ct}$. Primers are listed in Supplementary Table 1.

To extract viral RNA, after initial attachment of virus and inoculum removal, a sample of the freshly added media was collected as a 0 hpi supernatant. Following this, media was harvested and changed daily during infection such that each supernatant time point contained viral particles released over the prior 24 h. Viral RNA was extracted from 140 μl of each sample (2 samples per time point per experiment, though 0 hpi samples were only available from one experiment) using the QIAamp Viral RNA Mini Kit (Qiagen #52904) per manufacturer's protocol, and eluted in 35 μl nuclease-free water passed twice through the column.

qRT-PCR was performed using the universal EXPRESS One-Step SuperScript qPCR kit (ThermoFisher Scientific, Cat# 11781-200) with 2 μl input RNA. The CDC N1 primers and FAM-labeled probe were used for SARS-CoV-2 detection (https://www.cdc.gov/coronavirus/2019-ncov/lab/rt-pcr-panel-primer-probes.html) (IDT #10006713), and eukaryotic 18s rRNA primers and VIC-labeled probe (ThermoFisher Scientific #4319413E) were added to evaluate cellular RNA in the supernatant. For each sample, qRT-PCR was performed in technical triplicate using the ABI StepOnePlus™ Real Time System.

**Immunocytochemistry**. Cells grown on coverslips were fixed with 4% PFA for 10 min at room temperature. For permeabilization, cells were washed three times with PBS (Gibco, Cat# 14080-055) and incubated in PBST, which is 0.1% Triton X-100 (Sigma, Cat# T8787) 1× PBS solution, for 10 min at room temperature. Unspecific binding was blocked with 5% normal horse serum (Abcam, Cat# ab7484), 5% goat serum (Gibco, Cat# PCN5000), or 3% BSA (Sigma, Cat# A9647) in PBST for 1 h. Samples were then incubated overnight at 4°C with specific primary antibodies (Supplementary Table 2) diluted in 1% BSA in PBST, washed three times with PBS, and incubated with secondary antibodies (Supplementary Table 3) in PBST. Vectashield (Vector Laboratories, Cat# H1000) with DAPI (Cell Signaling, Cat# 4083) was used to mount the samples and visualize cellular nuclei. Negative control experiments without the primary antibodies were processed in parallel. Microscopy was performed using a Leica SP8 confocal microscope (Leica Microsystems) and Olympus FV1200 Laser Scanning Confocal Microscope. Microscopy images were analyzed using Fiji software. Percent of infected OPCs were quantified using 10× magnified images of three independent fields per condition. DAPI-stained nuclei were counted using automated cell counting on Fiji software and infected cells stained for SARS-CoV-2 nucleoprotein (NP) or dsRNA were counted manually.

**Immunohistochemistry**. Human vestibular tissue was fixed with 4% PFA for 4 h at room temperature. The fixed specimens were cryopreserved with a graded treatment of 10%, 20%, and 30% sucrose and then embedded in tissue freezing medium OCT compound. Frozen tissue blocks were sectioned into 12 μm cryosections on a Leica CM-1860 cryostat. For immunostaining, a 5% goat or horse serum in 0.3% Triton X-100 1× PBS solution was used for blocking, and a 1% BSA and 0.3% Triton X-100 1× PBS solution was used for primary/secondary antibody incubations.

**Statistical analysis and reproducibility**. All statistics were performed using GraphPad Prism 9.1.0 software and R version 3.5.1. Statistical significance was determined using an unpaired $t$-test with Welch's correction for multiple comparisons to a control group. For the analysis of SARS-CoV-2 RNA in the supernatant of OPC, we averaged technical triplicate Ct values for each sample. Normalized and relative expression values were determined for each sample with and without normalization to cell-free 18s rRNA copies using standard relative quantification (RQ) calculations (from ΔΔCt and ΔCt values, respectively) in Microsoft Excel. Sample sizes and number of biological or technical replicates are indicated in each figure legend.

**Reporting summary**. Further information on research design is available in the Nature Research Reporting Summary linked to this article.

## Results

**Clinical characteristics of patients**. Adult patients who developed audiovestibular dysfunction within 3 weeks of COVID-19 diagnosis, confirmed by PCR or serum antibody testing, were

studied. We used this time point as a surrogate for the outer limit of presumed ongoing active infection, given the Center for Disease Control and Prevention suggests a patient may shed infectious virus up to 20 days after symptom onset. A total of ten patients (six male, four female) met these criteria (Supplementary Data 1). Their ages ranged from 22 to 72 years. Nine of the ten patients experienced common COVID-19 symptoms, such as fever, cough, dyspnea, and/or fatigue between 21 days before and 14 days after the onset of hearing loss, tinnitus, or vertigo. Detailed medical histories of each patient are provided in Supplementary Note 1.

**Audiovestibular symptoms in COVID-19 patients**. Audiovestibular symptoms were the presenting symptoms of COVID-19 in three patients (patients 1, 2, and 3); for one of them (patient 2), hearing loss and tinnitus were the only symptoms of COVID-19. Seven patients developed audiovestibular symptoms after manifesting other symptoms of COVID-19. Audiometric data show that all patients experienced SNHL ranging from mild to profound (Fig. 1a). The majority had profound (>90 dB hearing level) or severe (71–90 dB hearing level) hearing loss, except patient 7, who had moderate hearing loss (56–70 dB hearing level) and patient 9, who had mild high-frequency hearing loss (24–35 dB hearing level) during COVID-19 infection. Table 1 summarizes the otoacoustic emission test results for patients 5, 6, 7, 8, 9, and 10 who completed this testing. Otoacoustic emission testing is used to confirm the presence of functional outer hair cells (OHCs) or the loss of OHCs because the testing is based on recording the sound made by OHC contraction with frequency-specific stimulation[23]. While the three-times less numerous inner hair cells (IHCs) transmit virtually all acoustic information to the brain as they synapse with 90–95% of type I auditory nerve fibers[24], IHC function cannot yet be clinically measured.

Nine out of ten patients experienced tinnitus. Six patients (patients 3, 4, 5, 6, 7, and 10) experienced rotational or linear vertigo, which typically lasted several days. Audiovestibular symptoms were treated with oral (PO) prednisone burst and taper in seven patients, or a combination of PO prednisone and intratympanic (IT) dexamethasone in two patients. Complete recovery of hearing was observed in patient 1 (who received prednisone 60 mg PO for 7 days and dexamethasone 10 mg/ml IT once) and patient 7 (who received prednisone 60 mg PO daily, tapering by 10 mg every 3 days), with modest recovery seen in patient 8 (who received prednisone 60 mg PO for 7 days, then tapering by 10 mg daily). None of the patients experienced chronic vestibular dysfunction attributable to COVID-19. Patient 10, who initially experienced right aural pressure, tinnitus, and episodic true rotational vertigo, as well as bilateral SNHL, was the only subject who had vestibular testing after he recovered from his SARS-CoV-2 infection. All six rotational receptors had normal gain and no asymmetry as measured by video Head Impulse Test. His otolithic function was normal and without asymmetry as measured by cervical vestibular-evoked myogenic potentials. The ocular vestibular-evoked myogenic potentials were absent bilaterally, which is not unusual at his age.

All patients underwent brain MRI during COVID-19 infection to rule out retrocochlear pathology. All MRIs were normal except for patient 7, a patient with SNHL and true rotational vertigo whose vestibule, cochlear and vestibular nerves, geniculate ganglion, and facial nerve showed diffuse enhancement ipsilateral to the ear with SNHL (Fig. 1b), consistent with inflammation related to virus infection. These data suggest a direct correlation between hearing loss, as quantified by audiometric data, and COVID-19.

**SARS-CoV-2 infection in adult human inner ear tissue**. To determine whether genes relevant to SARS-CoV-2 entry are expressed in sensorineural tissue of human vestibular end organs, fresh inner ear tissue was collected during surgical labyrinthectomies performed on patients who did not have SARS-CoV-2 infection. In the tissue, we identified MYO7A+/SOX2+ vestibular hair cells innervated by TUBB3+ vestibular primary afferent ganglion neurons which are enveloped by Schwann cells, and supported by SOX2+ cells that anchor these hair cells into the sensory epithelium (Fig. 2a and Supplementary Fig. 1a). We performed qRT-PCR on this tissue and found elevated levels of angiotensin-converting enzyme 2 (ACE2), transmembrane protease serine 2 (TMPRSS2), and FURIN mRNAs relative to undifferentiated hiPSCs (Fig. 2b). hiPSCs were used as a control based on prior reports showing ACE2 mRNA expression in hiPSCs and human embryonic stem cells Lin, Z. et al.[25]. Next, fluorescence immunohistochemistry was performed to identify the specific cell types that express ACE2, TMPRSS2, and FURIN proteins in human vestibular tissue. Co-expression of these proteins was identified in MYO7A+ hair cells (Fig. 2c and Supplementary Fig. 1b), especially at their apical regions. ACE2 was not detected in TUBB3+ peripheral axons of vestibular neurons (Supplementary Fig. 1b). However, myelin basic protein (MBP)-positive Schwann cells expressed ACE2 (Fig. 2d). Together, these results demonstrate that vestibular hair cells and Schwann cells are potential targets of SARS-CoV-2 in human peripheral vestibular organs.

To confirm that the expression of ACE2, TMPRSS2, and FURIN in adult human inner ear tissue facilitates viral infection, we exposed explanted human vestibular tissue to conditioned media ("mock", Fig. 2e–g) or SARS-CoV-2 (Fig. 2h–j). At 48 hpi, we observed SARS-CoV-2-infected hair cells (Fig. 2h, i) and Schwann cells (Fig. 2j), as assessed by the staining of viral double-stranded RNA (dsRNA, an intermediate product of coronavirus genome replication) and the staining of viral NP.

**2D models of the inner ear**. Because cells from a healthy, living human inner ear are typically inaccessible to researchers, we utilized hiPSC-derived OPCs and SCPs as in vitro 2D models of disease pathogenesis. Starting from human fibroblasts, reprogramming factors were used to restore pluripotency to somatic cells and generate person-specific hiPSCs (Supplementary Fig. 2), which were then differentiated into OPCs[18] and SCPs[19] using reliable and scalable protocols (Fig. 3a, b). The identity of OPCs was confirmed with otic lineage markers, such as GATA binding protein 3 (GATA3), paired box gene 2 (PAX2), paired box gene 8 (PAX8), and cochlin (COCH) (Supplementary Fig. 3a, b). The identity of SCPs was also confirmed with Schwann cell lineage markers such as nerve growth factor receptor (NGFR), SRY-box transcription factor 10 (SOX10), cadherin 19 (CDH19), growth-associated protein 43 (GAP43), and myelin protein zero (MPZ) (Supplementary Fig. 3c, d). To determine whether OPCs and SCPs express SARS-CoV-2 entry-related genes, qRT-PCR and immunocytochemistry were performed (Fig. 3c, d). ACE2 mRNA levels were higher in OPCs and in SCPs than in undifferentiated hiPSCs while TMPRSS2 mRNA expression was lower in the OPCs and SCPs than in undifferentiated hiPSCs (Fig. 3c). The relatively high level of TMPRSS2 expression in hiPSCs is consistent with its role in chromosomal segregation[26] (Supplementary Fig. 4). FURIN mRNA levels were higher in OPCs than in undifferentiated hiPSC, but their levels were not elevated in SCPs (Fig. 3c). For all three host entry factors studied, mRNA levels were higher in OPCs than in SCPs. Immunocytochemistry data suggest that all three proteins are expressed in hiPSCs. In hiPSCs, ACE2 was exclusively localized to the nucleus, TMPRSS2 was localized to the cytoplasm, and FURIN was mainly localized to the nucleus, with a minor presence in the cytoplasm and on the

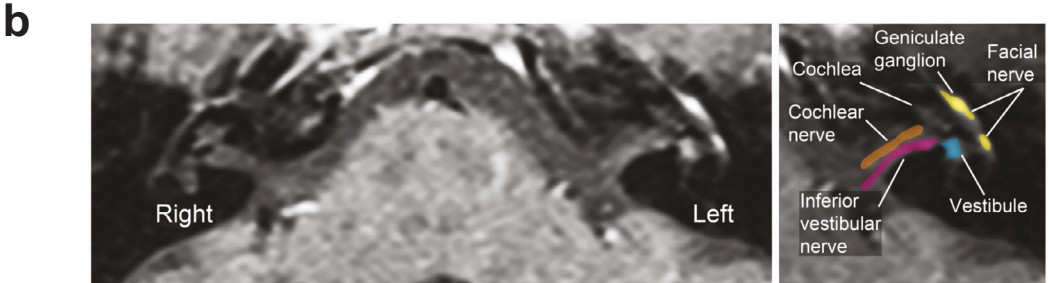

cell surface (Fig. 3d). In contrast, in OPCs, ACE2 and FURIN were found in the cytoplasm and cell surface (Fig. 3d). The distribution of ACE2 and FURIN in SCPs more closely resembled that in hiPSCs, with considerably less cytoplasmic/cell-surface expression of ACE2 and FURIN in these cells compared to OPCs (Fig. 3d). Finally, TMPRSS2 was detected in the cytoplasm in both OPCs and SCPs.

**SARS-CoV-2 infection in OPCs.** Having established that OPCs and SCPs express SARS-CoV-2 entry-related genes, we next infected the OPCs and SCPs with SARS-CoV-2. Addition of SARS-CoV-2 at a MOI of 1 to OPCs derived from hiPSC line UCSD112i-2-11 resulted in an infection rate of $26.01 \pm 4.94\%$ (mean $\pm$ SD) at 48 hpi as measured by percent of cells expressing viral NP (Fig. 4a). Infection rates were not dramatically increased

**Fig. 1 Clinical data. a** Audiometry during (1st row) and after (2nd row) SARS-CoV-2 infection. Audiograms in the 2nd row were performed 4 months (patient 1), 2.5 months (patient 2), 4 months (patient 3), 3.5 months (patient 4), 2 months (patient 5), 2 months (patient 6), 4 months (patient 7), 4 months (patient 8), 2 months (patient 9), and 3 months (patient 10) after COVID-19. Audiometric thresholds are defined relative to normal hearing level defined by the American National Standards Institute (ANSI, 2010). A masking dilemma is seen in patients 1, 4, 5, and 6 with profound SNHL unilaterally and normal hearing contralaterally; the air-conduction thresholds are the actual hearing levels for these patients. Right ear (red) air conduction O, bone conduction <. Left ear (blue) air conduction X, bone conduction >. CNT = could not test/no response. PTA = the average of pure-tone audiometric thresholds at 0.5, 1, 2, and 3 kHz. WR = word recognition score, defined as the percentage of spoken monosyllabic words discernable from a list typically read at 70 dB or at the level at which a patient's speech intelligibility curve plateaus. **b** Axial T1-weighted postcontrast MRI of the internal auditory canal of patient 7. Diffuse enhancement of the left geniculate ganglion, facial nerve, vestibule, cochlear, and vestibular nerves is seen. Facial nerve and geniculate ganglion (yellow), vestibule (blue), inferior vestibular nerve (purple), and cochlear nerve (orange) are colorized and superimposed over traditional images in the axial plane for the left ear.

| Table 1 Otoacoustic emission test results reflecting cochlear outer hair cell function. | | | | | | | | | | | | |
|---|---|---|---|---|---|---|---|---|---|---|---|---|
| **Patient** | **Hearing loss** | **Otoacoustic emission test results by frequency** | | | | | | | | | | |
| | | **Ear** | **2 kHz** | **2.5 kHz** | **3 kHz** | **3.5 kHz** | **4 kHz** | **5 kHz** | **6 kHz** | **7 kHz** | **7.5 kHz** | **8 kHz** |
| 5 | R profound SNHL | Right | Absent | Absent | Absent | Absent | Absent | Absent | Absent | Absent | Absent | Absent |
| | | Left | Present | Present | Present | Present | Present | Present | Present | Present | Present | Present |
| 6 | R profound SNHL | Right | Absent | Absent | Absent | Absent | Absent | Absent | Absent | Absent | Absent | Absent |
| | | Left | Present | Present | Present | Present | Present | Present | Present | Present | Present | Absent |
| 7 | L moderate low-frequency SNHL | Right | Present | Present | Present | Present | Present | Present | Present | Present | Present | Present |
| | | Left | Present | Present | Present | Present | Present | Present | Present | Present | Present | Present |
| 8 | B severe to profound SNHL | Right | Absent | Absent | Absent | Absent | Absent | Absent | Absent | Absent | Absent | Absent |
| | | Left | Absent | Absent | Absent | Absent | Absent | Absent | Absent | Absent | Absent | Absent |
| 9 | L mild high-frequency SNHL | Right | Present | Present | Present | Present | Present | Present | Present | Absent | Absent | Absent |
| | | Left | Present | Present | Present | Present | Present | Present | Present | Absent | Absent | Absent |
| 10 | B moderate SNHL | Right | Absent | Absent | Absent | Absent | Absent | Absent | Absent | Absent | Absent | Absent |
| | | Left | Absent | Absent | Absent | Absent | Absent | Absent | Absent | Absent | Absent | Absent |

*L* left, *R* right, *B* both, *SNHL* sensorineural hearing loss.

at a supraphysiologic MOI of 5, possibly reflecting activation of innate immune responses in the cells or the formation of defective interfering viral particles at high MOI. We also observed infection in OPCs derived from another hiPSC line, SK8-A (Supplementary Fig. 5). However, we observed only rare instances of infected SCPs—rates were considerably lower in SCPs than in isogenic OPCs infected with an equivalent inoculum of virus (<1% of SCPs were infected at a MOI of 1, compared to 7 ± 1.9% of isogenic OPCs) (Fig. 4b and supplementary Fig. 5). SARS-CoV-2 NP and dsRNA were both detectable in infected OPCs by 24 hpi (Fig. 4c). Detection of cells containing dsRNA dropped at 72 hpi, potentially reflecting the transient nature of this intermediate (Fig. 4d). Infected OPCs expressed ACE2 at levels subjectively similar to uninfected cells. We used viral RNA in cell supernatant—detected by qRT-PCR for the SARS-CoV-2 Nucleocapsid (N) gene[27]—as a surrogate to further quantify kinetics of viral replication and release. SARS-CoV-2 RNA was continuously released over the course of 72 h, with a burst in viral production between 24 and 48 hpi (Fig. 4e). The results were similar with and without normalization to 18s rRNA abundance in the supernatant (used as a control for RNA extraction efficiency, noting this value may be confounded by both division and death of cultured cells) (Supplementary Fig. 6). SARS-CoV-2 virus infection at a MOI of 1 did not affect measured viability of OPCs through 72 hpi (Fig. 4f). Taken together, these data confirm that OPCs and SCPs express SARS-CoV-2 entry-related RNAs and proteins, and that SARS-CoV-2 proteins are produced during virus replication.

**SARS-CoV-2 infection in 3D inner ear organoids**. To better understand SARS-CoV-2 infection within the inner ear, we utilized human inner ear organoids (Fig. 5a). These organoids are 3D tissue structures generated from hiPSCs that self-organize and mimic the architecture of multiple cell types of the inner ear, both morphologically and functionally. Notably, the expression pattern of ACE2 protein observed in inner ear organoids was similar to adult human vestibular tissue. ACE2 was detected in MYO7A+ hair cell-like cells, but not TUBB3+ neurites of neurons, which innervate hair cell-like cells (Fig. 5b). On the other hand, neuronal cell bodies expressed ACE2, mainly in the nucleus (Fig. 5c). In addition, ACE2 expression was also detected in Schwann cell-like cells expressing MBP (Fig. 5d). To model infection in this system, we inoculated human inner ear organoids with SARS-CoV-2. Organoids were harvested at 72 hpi for immunofluorescence microscopy. In keeping with the 2D culture model studies and infections of human vestibular explants, we observed viral infection in this 3D tissue culture model (Fig. 5e–h). Immunostaining for dsRNA revealed that the virus targeted vestibular hair cell-like cells (Fig. 5f, g). We also observed staining for dsRNA in MAP2-expressing neurons within the less highly structured areas of the organoids (Fig. 5h).

**Expression of SARS-CoV-2 cell entry-related genes in mouse inner ear**. To further investigate the distribution of SARS-CoV-2 cell entry genes in the auditory system, we turned to the mouse, because human cochlear sections are not readily available for immunohistochemistry. In the 6 weeks old CBA/CaJ murine cochlea, ACE2, TMPRSS2, and FURIN were expressed in the OC, spiral ganglion neurons (SGNs), and stria vascularis (SV, Fig. 6a). Cells within the vestibule that expressed all three proteins were located in the saccule and utricle (Fig. 6b). Our findings in the mouse cochlea and vestibule are in line with the published reports of ACE2, TMPRSS2, and FURIN expression in 11 weeks old male ICR mice[28]. These data demonstrate that, as in humans, SARS-CoV-2 entry-related genes are expressed in the mouse inner ear,

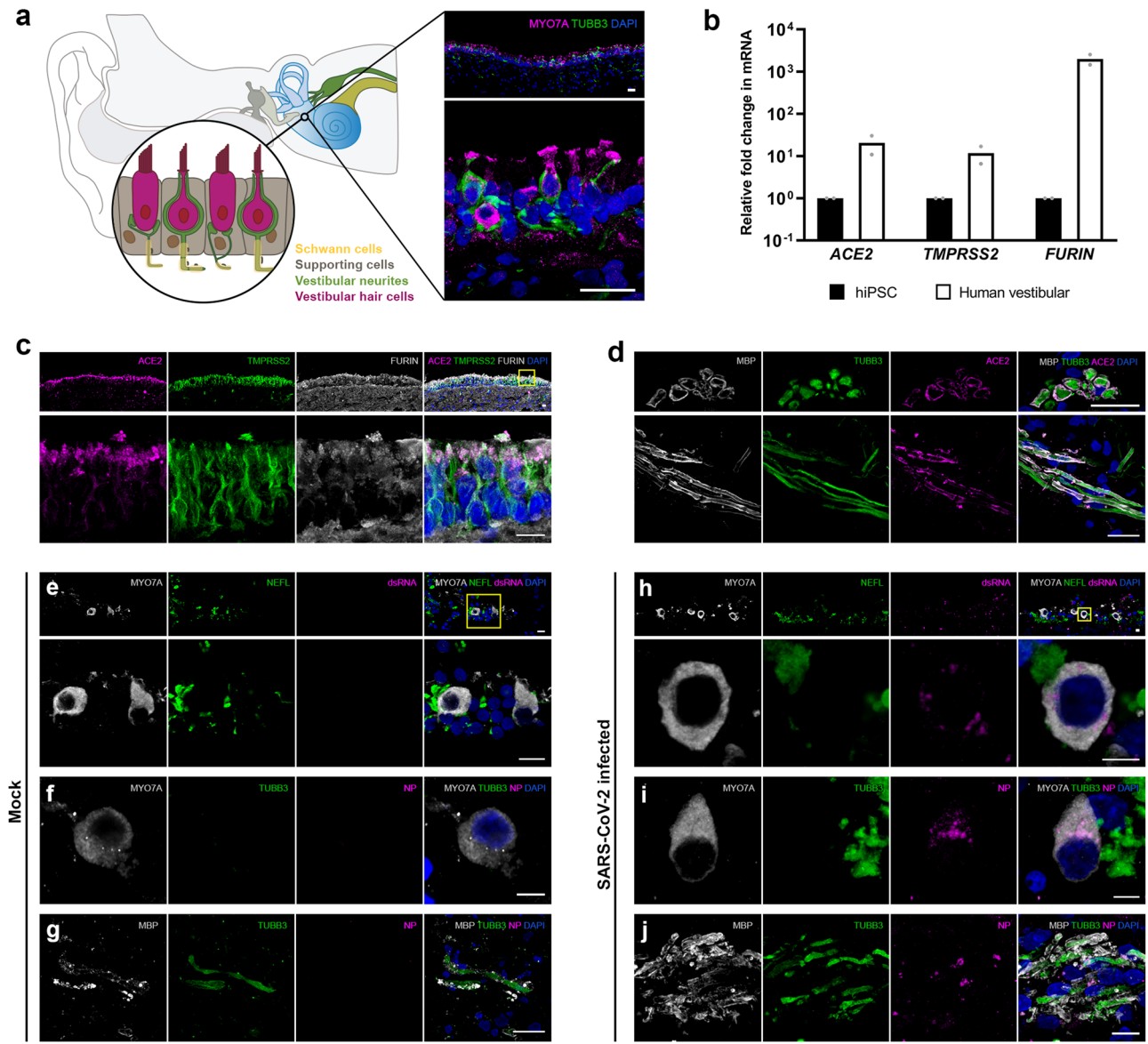

**Fig. 2 Expression of SARS-CoV-2 cell entry-related receptor and entry-associated proteases in human peripheral vestibular tissue. a** Surgical specimens of human vestibular end organs from the inner ear include vestibular hair cells expressing MYO7A, supporting cells, Schwann cells, and vestibular neurons expressing TUBB3. Scale bar = 20 μm. **b** mRNA expression of *ACE2*, *TMPRSS2*, and *FURIN* in human vestibular tissue relative to expression in hiPSCs. Individual gene expression levels in hiPSCs were set to 1. *N* = 2 biological replicates, 3 technical replicates; **P < 0.01, ****P < 0.0001; mean ± SEM. **c** Protein co-expression of ACE2, TMPRSS2, and FURIN in hair cells. Nuclei are stained with DAPI. In the overview of upper images, the rectangle is shown at a higher magnification. Scale bars = 10 μm. Representative images are from 2 biological replicates and at least 5 technical replicates with similar results. **d** Protein expression of ACE2 in Schwann cells. MBP is Schwann cell marker. Upper images show the cross-sectioned Schwann cells expressing ACE2. Representative images from 2 independent experiments with similar results. Scale bars = 20 μm. **e–j** Immunofluorescent staining of mock-infected (**e–g**) and SARS-CoV-2-infected human vestibular tissue (**h–j**). J2 staining of viral double-stranded RNA (dsRNA) and viral nucleoprotein (NP) expression in hair cells, Schwann cells, and not in neurons. Representative images from 2 biological replicates. Scale bars = 5 μm (**f, h, i**), 10 μm (**e, j**), 20 μm (**g**).

and levels of murine protein expression vary among different cell types. They also suggest that mouse models may be useful for future studies characterizing inner ear infection and its effects on hearing function.

## Discussion
To the best of our knowledge, this is the first report describing potential molecular mechanisms underlying audiovestibular dysfunction in patients with COVID-19 alongside a detailed description of audiovestibular symptoms in ten patients infected

with SARS-CoV-2. The temporal relationship between symptom onset and positive COVID-19 testing implicates SARS-CoV-2 infection as the cause of audiovestibular symptoms. None of our patients had evidence of middle ear infection on otologic examination or radiologic imaging, while all of our patients exhibited SNHL, suggesting direct viral infection or inflammation of the inner ear and cochleovestibular nerve as the cause of patients' symptoms. Indeed, one patient had MRI evidence of inflammation involving the inner ear, and cochlear and vestibular nerves (Fig. 1b). The lack of MRI findings in the majority of patients is in line with other viral etiologies of labyrinthitis that do not

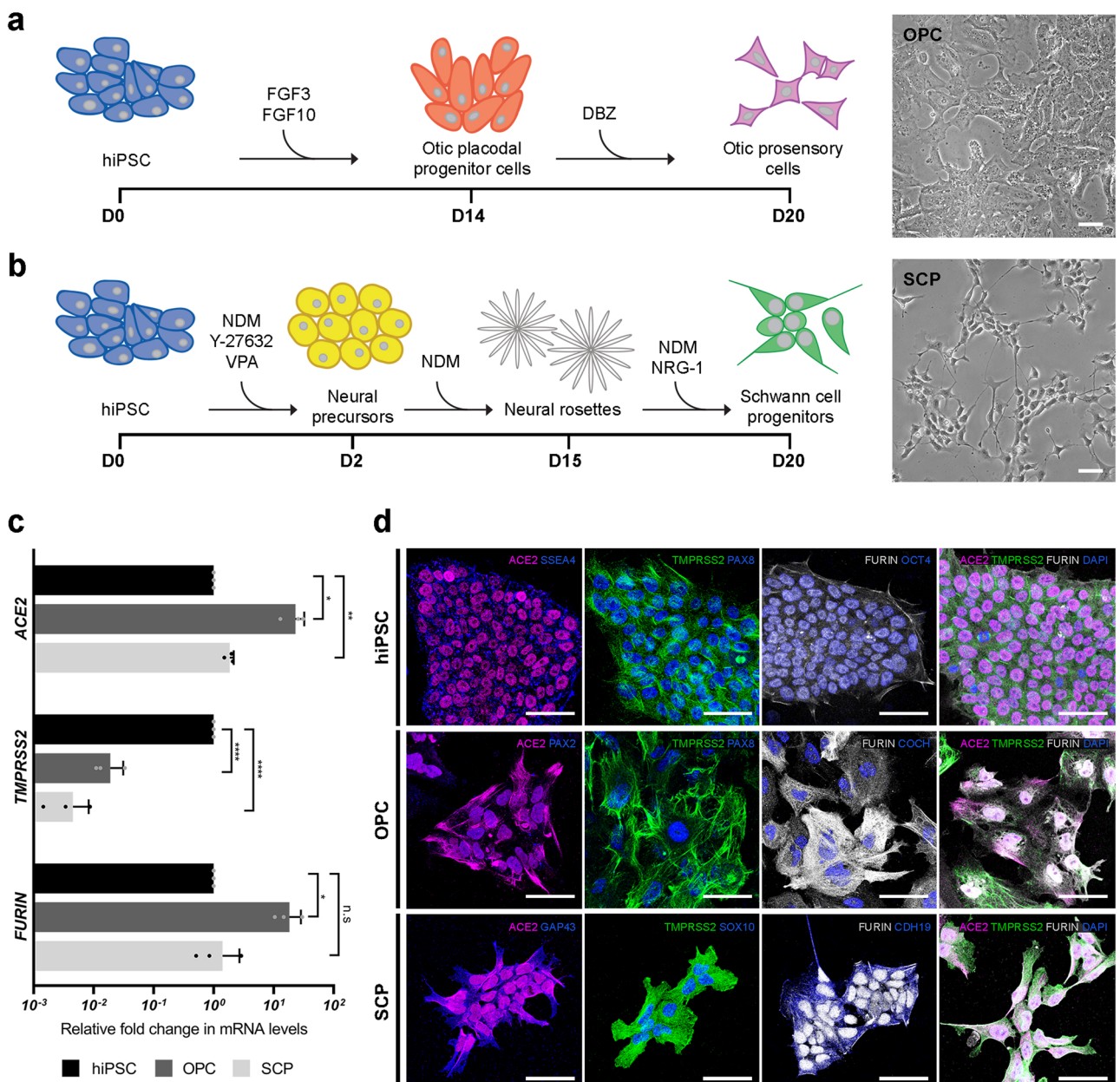

**Fig. 3 Differentiation of hiPSCs into OPCs and SCPs and confirmation of SARS-CoV-2 cell entry-related protein expression in these cell types.**
**a**, **b** Scheme illustrating the generation of OPCs and SCPs from hiPSC and a bright-field view of them. Scale bars = 50 μm. **c** Expression of *ACE2*, *TMPRSS2*, and *FURIN* mRNA in OPCs and SCPs differentiated from hiPSC line UCSD112i-2-11 relative to hiPSCs. Individual gene expression levels in hiPSCs are set to 1. $N = 3$ biological replicates, 3 technical replicates; n.s. > 0.05, *$P < 0.05$, **$P < 0.01$, ****$P < 0.0001$; mean ± SEM. **d** Expression of SARS-CoV-2 entry proteins together with pluripotency markers (SSEA4, OCT4, NANOG) in from hiPSC line UCSD112i-2-11, otic markers (PAX2, PAX8, COCH) in OPCs, and Schwann cell precursor markers (GAP43, SOX10, CDH19) in SCPs. Data are representative images from at least 3 independent differentiations. Scale bars = 50 μm.

frequently cause detectable abnormalities on clinical MRI[29,30]. However, when inner ear or nerve enhancement is seen on T1-weighted MRI, it is highly specific and often correlates with objective signs and subjective symptoms. Importantly, MRI in all patients ruled out other retrocochlear pathology, such as vestibular schwannoma, stroke, or intracerebral hemorrhage as potential causes of SNHL or vertigo. The rate of complete hearing recovery we witnessed in two of the ten patients is consistent with the rate (1 in 3) of spontaneous recovery in patients diagnosed with sudden idiopathic SNHL prior to the COVID-19 pandemic[31,32]. This suggests that COVID-19-related hearing loss may be heterogenous in mechanism or severity, and that pathogenesis may be shared across multiple etiologies of SNHL.

The clinical portion of this study is limited by its small size and unknown prevalence of audiovestibular dysfunction in COVID-19, because universal testing has not been adopted during the pandemic. Audiovestibular symptoms may not have been documented in patients with more severe morbidity or in those succumbing to their infection; moreover, there is the possibility that patients with sudden isolated SNHL or mild vestibular symptoms may have avoided in-person evaluation because of concerns about becoming infected with SARS-CoV-2. An additional limitation is the lack of histopathologic examination of human inner ears from COVID-19 patients to further evaluate different cellular mechanisms of audiovestibular dysfunction. Routine tissue biopsy of the living human inner ear is not possible due to the organ's

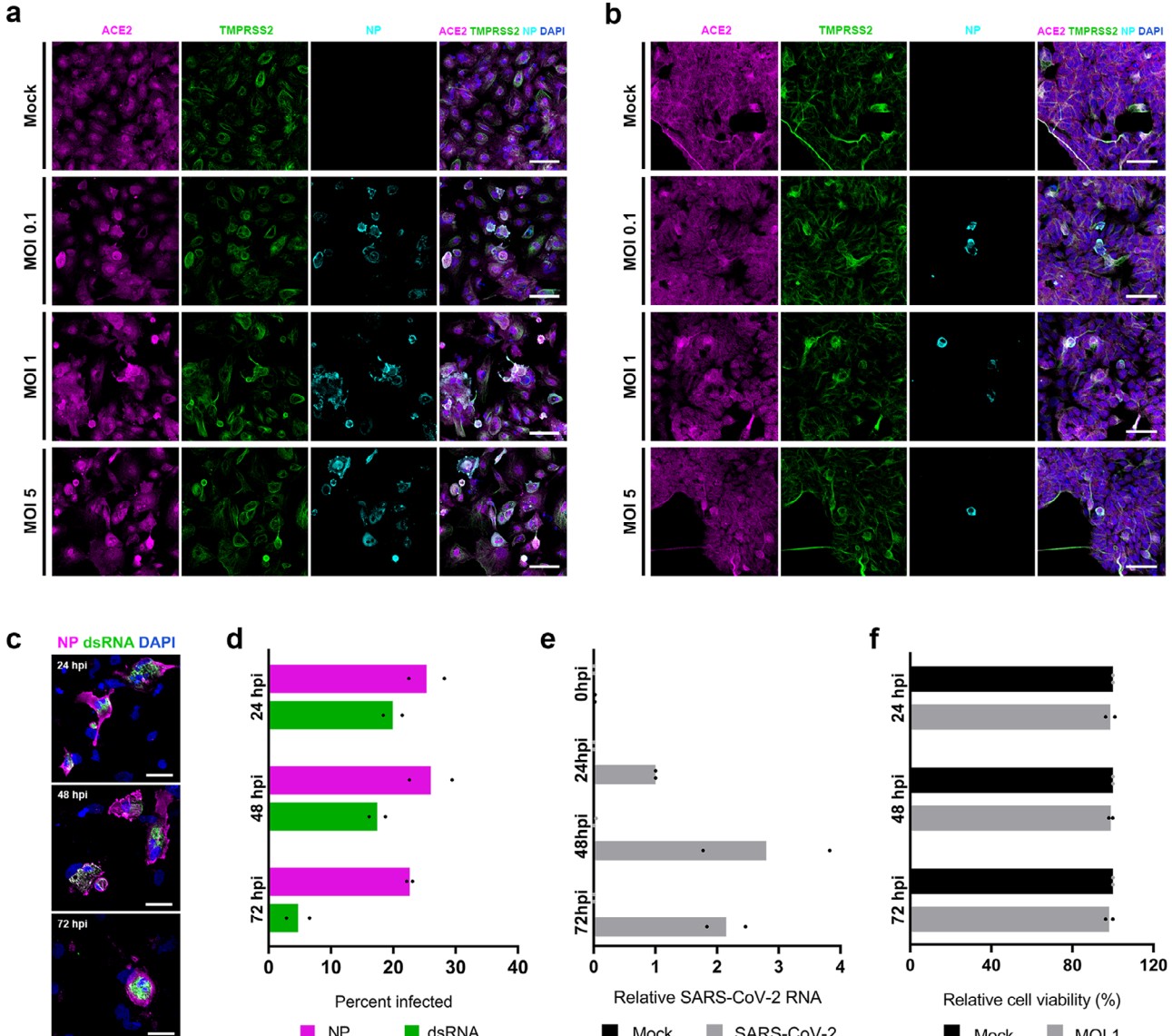

**Fig. 4 SARS-CoV-2 infection of OPCs and SCPs generated from hiPSC. a** Infected OPCs differentiated from hiPSC line UCSD112i-2-11 were visualized by immunostaining for viral nucleoprotein (NP) and co-immunostaining for ACE2 and TMPRSS2 at 48 hpi. Representative images from 4 independent experiments with similar results. Scale bars = 100 μm. **b** Infected SCPs differentiated from hiPSC line SK8-A are visualized by immunostaining for viral NP and co-immunostaining for ACE2 and TMPRSS2. Note that these images are chosen to show infected cells and are not representative fields of the overall culture of cells. Similar results were seen in 2 biological replicates each with 2 nested technical replicates. Scale bars = 50 μm. **c** SARS-CoV-2 NP and dsRNA stained with the J2 antibody were detectable in infected OPCs at MOI 1 at 24, 48, and 72 hpi. Data are representative images from 2 independent infections. Scale bars = 50 μm. **d** Quantification of data shown in (**c**). Mean percentages of infected OPCs are plotted as detected by immunostaining with either anti-NP or J2 antibody at 24, 48, and 72 hpi. Points represent the mean value from each of two independent experiments (*N* = 2, each with 3 nested replicates). MOI = 1. **e** Mean fold change of SARS-CoV-2 RNA in the supernatant of OPCs compared to 24 hpi samples in each experiment (MOI = 1). Except for the zero hour sample, samples harvested at each time point represent viral RNA released over previous 24 h only. Bar height represents geometric mean of two experiments, each with two nested replicates. Points represent the mean value from each of two independent experiments (*N* = 2, each with 2 nested replicates). Normalization to a cell-associated housekeeping gene RNA was not performed. **f** OPC cell viability was measured using a bioluminescence assay for ATP. Mean relative light units (RLU) for each sample is plotted as a percentage of RLU in the equivalent mock-infected sample at 24, 48, and 72 hpi. *N* = 2, 3 technical replicates. OPC infections were performed at a MOI of 1.

small size, complex 3D anatomy, and encasement in the densest bone in the body; therefore, histopathologic examination can only be done on autopsy specimens[33]. While there are no published reports of inner ear histopathology in patients who died of COVID-19, SARS-CoV-2 has been identified in the middle ear and mastoid bone from COVID-19-positive decedents[34].

Given these limitations, we developed the first human in vitro 2D and 3D models of SARS-CoV-2 otic infection derived from hiPSCs, and compared data from these samples to difficult-to-

obtain explants of adult human inner ear samples. Hair cells appear to be infected in all human inner ear tissue models examined. This is consistent with our finding that ACE2, TMPRSS2, and FURIN are expressed in these cells in subcellular locations predicted to be accessible to viral spike proteins (present in the cytoplasm and/or at the cell surface). While hair cells studied in our cellular models and inner ear explants were of vestibular origin, we expect that cochlear hair cells can also be infected with SARS-CoV-2 because vestibular and cochlear hair

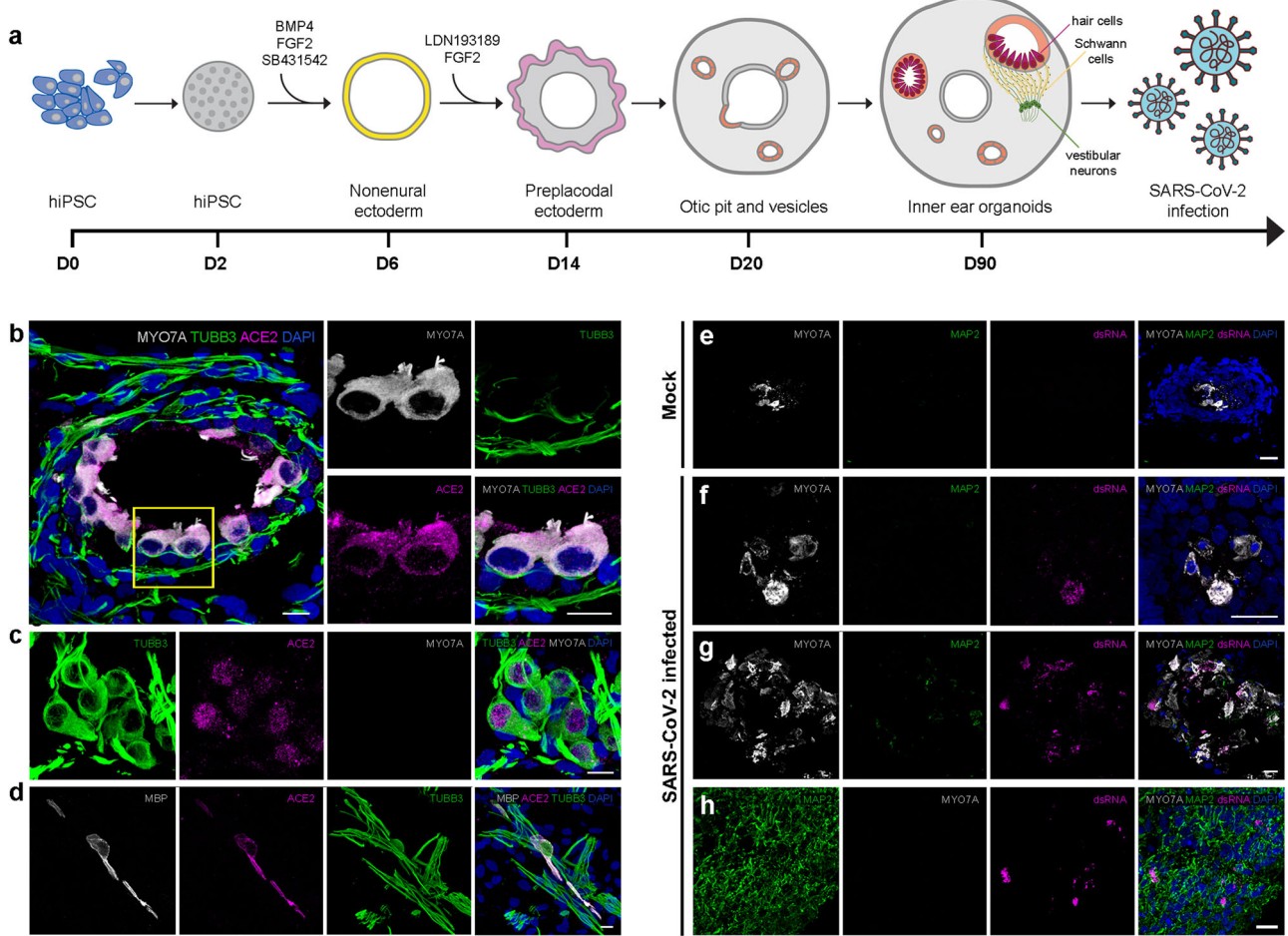

**Fig. 5 Hair cell-like cells and neurons from inner ear organoids infected with SARS-CoV-2. a** Schematic overview of the generation of inner ear organoid from hiPSCs and infection with SARS-CoV-2. **b–d** Protein expression of ACE2 in inner ear vestibular organoids containing MYO7A+ hair cells (**b**), TUBB3+ cell body of ganglion neurons (**c**), and MBP+ Schwann cells (**d**) generated from hiPSC line SK8-A. Scale bars = 10 µm. **e–h** Immunofluorescent staining of mock (**e**) and SARS-CoV-2-infected differentiated organoid culture. Viral dsRNA, stained by the J2 antibody, colocalized with the hair cell marker MYO7A (shown in (**f**) and (**g**)) and neuronal marker MAP2 (shown in (**h**)). Scale bars = 20 µm. Representative images from at least 3 independent organoids.

cells share important anatomic and physiologic similarities[32] and we noted the expression of SARS-CoV-2 entry genes in mouse hair cells, and our patients experienced both vestibular and auditory symptoms.

By contrast, we have inconsistent data that Schwann cells are infected. SCPs were not readily infected in 2D culture. This is in keeping with the finding that there is relatively low expression of SARS-CoV-2 entry cofactors in SCPs. Moreover, in SCPs, ACE2 and FURIN are located in the nucleus of cells, where they may not be accessible to the entering virus. However, infected mature Schwann cells were observed in human explants. Further evaluation of cofactor expression levels and subcellular location of these proteins in the mature Schwann cells may explain differences between the immature SCPs and mature cells in the adult human explants. It should be noted that alternative routes of entry have been described for SARS-CoV-2, and these may also be accessible in mature Schwann cells. The lack of ACE2 expression in mouse cochlear Schwann cells is suggestive of either heterogeneity in Schwann cell protein expression, or differences between human and mouse Schwann cells, underscoring the importance of using human tissue models for the study of infectious diseases.

Finally, in one model of the inner ear organoids, we observed neurons infected by SARs-CoV-2. However, this is in a much less structured area of the organoid that might not recapitulate adult human tissue. We did not observe infection in neurites of explanted tissue, and similarly, ACE2 expression was not observed in neurites.

Combined, these data suggest that hair cells may be particularly vulnerable to infection by SARS-CoV-2 in the human inner ear, and they provide a mechanistic explanation for the tropism—namely that known cofactors for infection are present in virus-accessible locations within the hair cells, whereas they are less available to the entering virus in other cell types. Our work establishes 2D and 3D models for the study of human audiovestibular disease. Since 3D models reconstitute the accurate and complex microenvironment found in living organisms, such as cell–cell junctions and apical–basal polarity[35], they may serve as better in vitro models for hearing loss and balance disorders. For this, further optimization of the protocol is necessary to improve differentiation efficiency[34]. Together, these 2D and 3D models may serve as platforms for future development and testing of novel vaccines and therapeutics with otoprotective and otoregenerative properties.

Our data propose at least three possible mechanisms for the acute audiovestibular symptoms observed in COVID-19 patients. First, the virus may directly infect and kill cochlear hair cells (which detect sound), vestibular hair cells (which detect head movement, acceleration and deceleration), and/or primary afferent auditory and/or vestibular nerve cells (which relay information to the brain). While viability studies did not show cell death

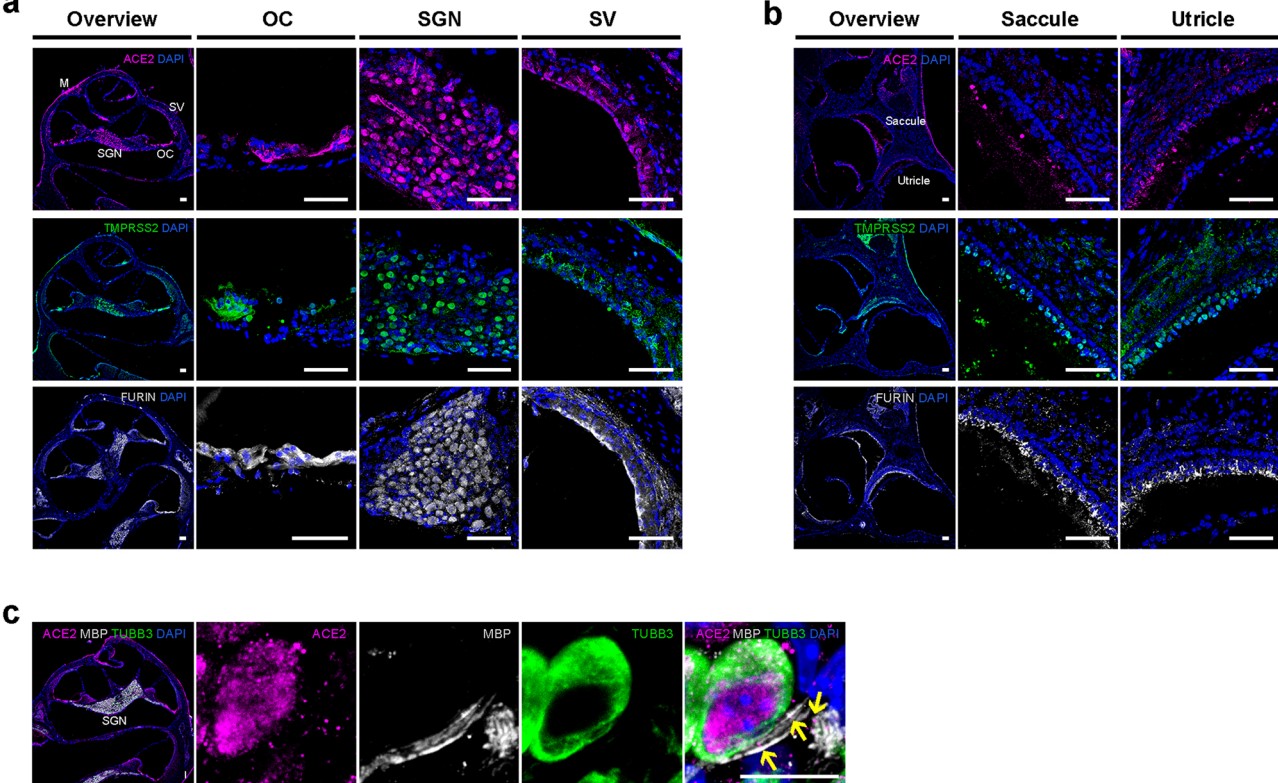

**Fig. 6 Expression of SARS-CoV-2 cell entry-related receptor and entry-associated proteases in different cell types of the mouse inner ear. a, b** The presence of ACE2, TMPRSS2, and FURIN in the mouse cochlea (**a**) and vestibule (**b**). In the cochlea, all 3 protein-positive cells are located in the OC, which contains sensory hair cells and supporting cells, the SGNs, which innervate hair cells, and the SV, which contains numerous blood vessels. In the vestibule, the cells co-expressing all 3 proteins are located in the saccule and utricle. Data are representative images from at least 3 independent experiments. Scale bars = 50 μm. **c** Small expression of ACE2 (indicated by yellow arrows) in MBP-expressing Schwann cells. Data are representative images from 3 independent experiments. Scale bars = 10 μm.

(Fig. 4f), more subtle cytopathic effects may occur, perhaps over a longer time frame than studied in this report.

Relevantly, the absence of OHC function in patients 5, 6, 8, and 10, as measured by otoacoustic emissions, correlated with the SNHL, suggesting that this largely irreversible loss was due to SARS-CoV-2 infection of the auditory hair cells. For patient 9, her otoacoustic emission testing showed a loss of OHC function between 3 and 8 kHz on the left and symptomatic side, and between 7 and 8 kHz on the asymptomatic side, suggesting that she had a preexisting damage of OHCs bilaterally between 7 and 8 kHz and that her SARS-CoV-2 infection resulted in additional left-sided OHC injury in the region of 3–6 kHz. Transient cell dysfunction, rather than cell death, is an alternative mechanism given the presumed low incidence of permanent audiovestibular dysfunction due to COVID-19, and the fact that two of our cases experienced transient hearing loss and six experienced transient vertigo or imbalance. For example, while patient 10's otoacoustic emission testing was consistent with bilateral OHC injury, his vestibular testing after he recovered from COVID-19 was essentially normal. This suggests that if the vestibular hair cells were infected with SARS-CoV-2, they have recovered; it has been demonstrated that limited spontaneous regeneration of vestibular hair cells can occur after damage in mature mammals[36]. Alternatively, it is possible that his vestibular hair cells were not infected and that his symptoms were due to a different mechanism, such as endolymphatic hydrops as elaborated below. Second, direct viral infection of inner ear cells could trigger innate interferon-mediated responses, followed by release of pro-inflammatory cytokines, and the resulting inflammatory response may damage inner ear cells, leading to hearing loss, tinnitus, and vertigo. Third, damage to the stria vascularis may disrupt the endocochlear potential and potassium homeostasis of the endolymph within the cochlear duct, leading to hearing loss.

Likewise, there are several potential paths by which SARS-CoV-2 might directly access the inner ear (Fig. 7). The first is through the central nervous system, which it may enter via the olfactory groove. COVID-19 patients can experience ageusia, anosmia, and central nervous system dysfunction[37], including trochlear nerve dysfunction accompanied by trochlear nerve enhancement on MRI[38]. In our series, three of the ten (30%) patients had evidence of central nervous system entry of the virus; two of the ten (20%) patients had dysgeusia and anosmia and one had cranial nerve enhancement of the cochlear, vestibular, and facial nerves on the side of the patient's hearing loss on contrast-enhanced MRI, which has not been previously reported (Fig. 1b and Supplementary Data 1). In these patients, SARS-CoV-2 potentially seeded cerebrospinal fluid, enabling delivery to the cranial nerves, and subsequently the inner ear. A second path is through the endolymphatic sac, which is the immunologic interface between the inner ear and the rest of the body[39]. Two of the ten patients (patients 1 and 7) had reversible SNHL. Clinically, their pattern of hearing loss and the recovery, as well as the presence of otoacoustic emissions in patient 7 who had them measured, is consistent with endolymphatic hydrops, suggesting that the endolymphatic sac was the site of entry of SARS-CoV-2 in these patients. A third proposed route is hematogenous spread, through the stria vascularis due to virally induced compromise of the blood–labyrinthine barrier. Systemic inflammation is known to increase vascular leakage of the stria vascularis[40], which could

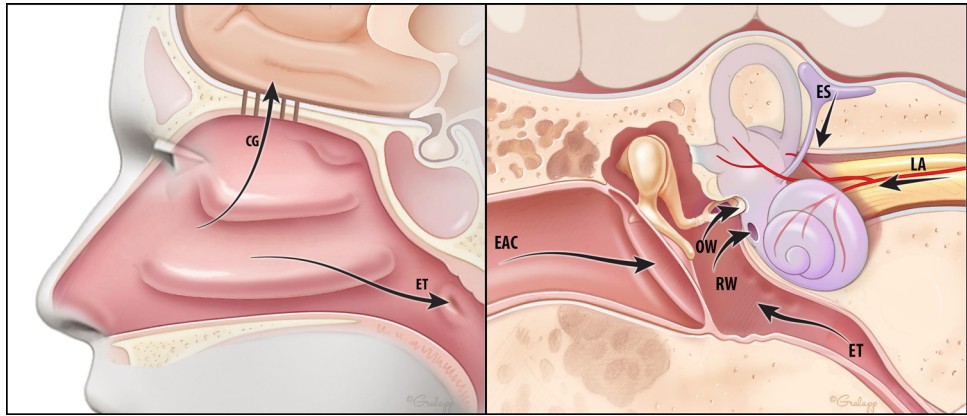

**Fig. 7 Potential paths for SARS-CoV-2 entry into the inner ear.** Arrows indicate potential paths via the nose and olfactory foramina (OF) into the central nervous system; via the endolymphatic sac (ES); via labyrinthine artery (LA) to ultimately reach stria vascularis; via round window (RW) and oval window (OW) membranes which the virus could reach through the Eustachian tube (ET) or external auditory canal (EAC), middle ear and mastoid. The diagram within this figure was drawn by Chris Gralapp and is being reproduced with permission.

allow entry of the virus and pro-inflammatory cytokines into the inner ear. This may have been the route of entry for patient 4 who required ventilator support due to presumed systemic cytokine storm. Since blood–labyrinthine and blood–brain barrier share structural and functional similarities, it is relevant that SARS-CoV-2 can damage the blood–brain barrier in a human cellular model in vitro[41]. Finally, in a fourth path, SARS-CoV-2 may traverse round or oval window membrane. SARS-CoV-2 has been shown to infect the mucosa of the middle ear and mastoid, supporting this possibility[34]. However, the lack of middle ear effusion or infection in the described patients suggests this pathway is less likely.

We note that we observed in vitro SARS-CoV-2 infection of OPCs, a cell type present in the developing fetal inner ear. While pediatric populations have been largely spared of COVID-19 symptoms and maternal–fetal transmission of SARS-CoV-2 is thought to be rare[42], the potential tropism of SARS-CoV-2 for OPCs raises concerns. The developing inner ear is notoriously sensitive to congenital viral infection: congenital cytomegalovirus (CMV) accounts for up to 30% of congenital hearing loss[43]; vertical transmission of rubella virus, now thankfully rare, was once another major cause of congenital deafness[44]; and case series also suggest a link between congenital Zika virus infection and hearing impairment[45]. Our human in vitro model of otic progenitor cell viral infection establishes a platform that can be used in future studies of other viruses known to cause developmental hearing loss, including that attributable to CMV and Zika virus. It may be reasonable for providers to monitor hearing more closely in children born during the COVID-19 pandemic.

Although continued research is necessary to understand SARS-CoV-2 pathogenesis and transmission in the inner ear, we begin to build on a causative link between SARS-CoV-2 and audio-vestibular dysfunction. Pursuing this relationship is important in understanding and ultimately combating the ever-growing list of COVID-19-related clinical manifestations. Based on the clinical, human in vitro, and animal evidence presented here, we advocate that health care providers monitor both for signs and symptoms of audiovestibular impairment in COVID-19 patients.

## Data availability
All data generated or analyzed during this study are included in this published article and its supplementary information files. Source data for the main figures of the manuscript is available as Supplementary Data 2. All other data are available from the corresponding author on reasonable request.

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

## Acknowledgements
We thank Sachiyo Katsumi, Masaharu Sakagami, and Xiaojie Ma for assistance with experiments, Jessica E. Sagers for help with preliminary analysis of SK8-A iPSCs, the Animal Care Facility at Massachusetts Eye and Ear for mouse-colony maintenance, the Ragon BSL3 management team for experimental support and facility maintenance, and the reported patients for use of their clinical information. This work was supported by the National Institutes of Health grants R01 DC015824 (K.M.S.) and U19AI131135 (L.G.), as well as Nancy Sayles Day Foundation, the Barnes Foundation, the Remondi Foundation, Sheldon and Dorothea Buckler and Bertarelli Foundation endowed professorship (K.M.S.).

## Author contribution
K.M.S. and L.G. conceived the project, designed the experiments, analyzed data, and jointly supervised all aspects of the research. M.J. helped design the study, performed hiPSC differentiation into OPCs and inner ear organoids, immunohistochemistry, immunocytochemistry, imaging, and qRT-PCR. D.H. developed the SCP protocol, performed hiPSC differentiation into SCPs, and helped with immunohistochemistry. K.O. helped design infectious studies, performed immunostaining of infected samples and performed and analyzed viral RNA extraction and qRT-PCR. Y.Z. and V.L. conducted infection of human cells with SARS-CoV-2. Y.Z. performed microscopy on infected samples. A.B. helped with specimen collection, RNA extraction, and tissue preparation. P.A.W., A.V., T.K., R.C., and N.S. provided clinical histories, MRI images, and data on patients with COVID-19. F.S., D.W., and K.M.S. provided human surgical inner ear samples. C.M. helped with tissue processing. All authors analyzed the data. M.J. and K.M.S. wrote the paper, with editorial input from all authors. All authors approved the final version of the manuscript.

## Competing interests
The authors declare no competing interests.
