## [Peer Review File. · Communications Medicine]

Reviewers' comments:

Reviewer #1 (Remarks to the Author):

The authors present the clinical data of a multicenter study on involvement of the inner ear with SARS-CoV-2 infection and associated audiovestibular symptoms. They report about the so far largest series of patients involving 10 with different degrees of hearing loss, tinnitus and vertigo. Some of these patients showed recovery of hearing loss and other symptoms with appropriate treatment of the underlying disease. MRI findings showed involvement of labyrinth in one patient and no retro-cochlear pathology indicating that the inner ear is directly affected.

In order to identify the pathogenesis, the expression of SARS-CoV-2 cell entry related genes and proteins in mouse inner ear tissue was investigated. It was a high expression found in vestibular cells. In addition human-induced pluripotent stem cells derived otic prosensory cells (OPCs) and Schwann cell precursors (SCPs) were infected with SARS-CoV-2 virus followed by an increase in the expression of virus-specific RNA and NC-proteins. Some cells also showed direct cytopathic effect. The authors conclude that the inner ear has direct targets for infection and discuss potential mechanisms with either, direct virus infection of the cochlear cells, indirect damage by induced inflammatory response or damage through the Stria vascularis with disruption of endocochlear potential. The author discussed potential paths of the virus into the inner ear with either through the central nervous system entering through the olfactory groove, the endolymphatic sac as the location for immune response in using endolymphatic hydrops, through the Stria vascularis due to virally induced damage of the blood labyrinth barrier or travers infection through round or oval window. The authors also discuss the limitations of their current study due to the small number of patients, the missing access to living human inner ear tissue of infected patients and the valuable clinical findings in these ten patients.

They also state that they have developed a first human in vitro model for SARS-CoV-2 infections of the inner ear with a platform that could serve for future development and testing of vaccines and therapeutics with otoprotective and otoregenerative properties. The derived cells are close to the developing inner ear. The findings might indicate specific sensitivity of the developing ear towards SARS-CoV-2 virus infection and suggest investigating also on audiovestibular impairment not only in adult patients but also in children with congenital hearing loss during the pandemic.

The supplementary information to this paper is very useful with specific comments on the used methods and the single patients.

The overall impression of the work is excellent. It is the so far best report on clinical findings of patients with SARS-CoV-2 infection and audiovestibular symptoms. The paper is the first one to show the specific target genes and proteins of the human inner ear for the virus and shows the potential possibility of virus infection with associated cytopathology of inner ear cells. The paper gives a very good basis for future clinical and experimental investigations and studies to identify the true mechanisms of hearing loss and the potential pathways of infection into the inner ear.

In detail the case reports show very different clinical pictures of the patients with proven SARS-CoV-2 virus infection. It would be very interesting to have further investigations on each of these patients such as audiometric testing including electrocochleography, the glycerol test, otoacoustic emissions as well as neurotological tests to specify the underlying pathophysiology of the clinical symptoms.

This would allow to differentiate also the therapy in each of these patients in order to restore hearing at least to a certain extend e.g. in patients with endolymphatic hydrops.

The paper is well written and organized. Illustrations are very illustrative. The legends of the figures are appropriate. The current literature is well reviewed and references are listed.

Overall, I would like to recommend to accept this paper with high priority for publication.

Reviewer #2 (Remarks to the Author):

The topic of this study is timely and it is striking that sensorineural hearing loss (SNHL) can be included in considerable neurological disorders in COVID-19 infection.

This study consists of clinical and experimental assessments, of which include various types of materials including human iPS or iPS-derived cells, human vestibular specimens, mouse cochleae and vestibular peripherals. The authors concluded that the hair cell is the target of COVID-19 based on fragile, not steady, results of various kinds of assessments. All the findings obtained from various materials are fragmental and superficial. No strong evidence showing that COVID-19 causes SNHL or hair cell damage is not obtained.

1) Audiograms of patients with COVID-19 infection are valuable, but it is difficult to explain that SNHL was caused by viral infection. The data may be suitable for case reports.

2) Use of human iPS cell-derived cells is challenging and interesting. The problem is the use of otic prosensory cells derived from human iPS cells, not hair cells or hair cell-like cells after differentiation. The authors described that the expression of ACE2 in otic prosensory cells indicates the vulnerability of hair cells. Otic sensory progenitors can differentiate into various types of inner ear cells. It is difficult to say the vulnerability of hair cells through the findings of otic prosensory cells. The authors should use hair cells or hair cell-like cells derived from human iPS cells.

3) An experiment of COVID-19 infection was conducted using otic progenitor cells. The use of mouse cochlear and/or vestibular explant cultures for this purpose will be more persuasive.

Reviewer #3 (Remarks to the Author):

Comments on "Direct SARS-CoV-2 infection of the human inner ear may underlie COVID-19-associated audiovestibular symptoms" by Jeong et al.

COVID-19, which is caused by the novel coronavirus SARS-CoV-2, mainly presents as a respiratory disease, however certain patients can develop multitude of non-respiratory symptoms due to viral replication in other organs/tissues. For instance, SARS-CoV-2 has the ability to spread to and replicate in tissues of the sensory system, thereby leading to symptoms such as (temporary) loss of taste or hearing difficulties. In the present manuscript Jeong and coworkers investigated COVID-19-associated audiovestibular symptoms in patients and further utilized human induced pluripotent

stem cell (hiPSC)-derived otic prosensory cells (OPCs) and Schwann cell precursors (SCPs) for expression of the critical SARS-CoV-2 host factors ACE2, TMPRSS2 and furin and their permissiveness to SARS-CoV-2 infection. Their main findings are, (i) OSCs and SCPs express ACE2, TMPRSS2 and furin; (ii) both OSCs and SCPs are susceptible to SARS-CoV-2 infection; (iii) infection of OSCs is more efficient compared to SCPs. Finally, the authors show that SARS-CoV-2 cell entry-related genes and proteins are also efficiently expressed in mouse inner ear tissue, suggesting that mice can be used as a model system for studying SARS-CoV-2 infection of the inner ear.

The manuscript covers a timely topic and the reported findings are of general interest. The experiments are well conducted and the findings support the conclusions. The text is well written. Altogether, this study is of high quality and, after taking care of the points specified below, can be endorsed for publication.

Specific points

- SARS-CoV-2 infection was exclusively studied by fluorescence microscopy (staining of viral nucleoprotein or dsRNA). Especially the data in Fig. 4 would benefit from viral growth kinetics (ideally with several timepoints between 1 and 96 hpi) in the respective cell models.
- Infection rates appear relatively low (Fig. 4A and B) and especially for SCPs only a couple NP-positive cells were detected. This could be due to the low titer inoculum (MOI = 1). The authors should also test higher amounts of virus (MOI = 3 or 10 in order to “hit” more cells).

Minor points

- The labeling of the panels presented in Fig. 4 does not match the figure legend and the main text. For instance, the data presented in panel C of the figure are labeled as panel C in the main text and the corresponding figure legend; panel E is not mentioned in the legend/main text.
- Please specify the statistical tests used (Fig. 3C Extended Fig. 2C and D, Extended Fig. 3A and C).
- Page 4: Please change “herpes family” and “paramyxoviridae” to “Herpesviridae” and “Paramyxoviridae”, respectively. Both terms should be italicized.
- Multiple pages: Please change “nucleocapsid” to “nucleoprotein” and the corresponding abbreviation from “NC” to “NP” (main text and figures).
- Multiple pages: Please change “SARS-COV-2” to “SARS-CoV-2” in the subheadings on pages 6 to 9.

AUTHOR RESPONSES

Reviewer #1

In detail the case reports show very different clinical pictures of the patients with proven SARS-CoV-2 virus infection. It would be very interesting to have further investigations on each of these patients such as audiometric testing including electrocochleography, the glycerol test, otoacoustic emissions as well as neurotological tests to specify the underlying pathophysiology of the clinical symptoms. This would allow to differentiate also the therapy in each of these patients in order to restore hearing at least to a certain extend e.g. in patients with endolymphatic hydrops.

Thank you for pointing out that there are a number of diagnostic studies that could potentially refine our understanding of the mechanisms underlying the observed audiovestibular clinical phenotypes. The revised manuscript includes new data on otoacoustic emission testing, which was completed in 6 out of 10 subjects (new Table 2). We have revised the results section accordingly (page 6, paragraph 1). In addition, one patient had vestibular testing after he recovered from his SARS-CoV-2 infection. This testing included video Head Impulse Testing, the ocular and vestibular evoked myogenic potential testing (page 6, paragraph 2). These new data suggest that SARS-CoV-2 infection impaired outer hair cells function because the majority of the tested patients had absent otoacoustic emissions in the symptomatic ears. These clinical data complement and extend our mechanistic studies showing that hair cells are most readily infected by SARS-CoV-2 in adult human inner ear explants as well as in human induced-pluripotent stem cell-derived otic

prosensory cells and otic organoids. The revised discussion is amended accordingly (page 14, paragraph 2 and page 15, paragraph 1).

While beyond the scope of the current study, our results strongly motivate additional clinical testing recommended by the reviewer in patients with COVID-19 and audiovestibular symptoms. Electrocochleography and the glycerol test do have potential for greater insight into whether a few of these subjects experienced endolymphatic hydrops; however, electrocochleography with significant hearing loss is not possible and very few centers in the world still perform the glycerol test.

The paper is well written and organized. Illustrations are very illustrative. The legends of the figures are appropriate. The current literature is well reviewed and references are listed.

Overall, I would like to recommend to accept this paper with high priority for publication.

We thank the reviewer for kind remarks and enthusiasm about our study.

Reviewer #2

1) Audiograms of patients with COVID-19 infection are valuable, but it is difficult to explain that SNHL was caused by viral infection. The data may be suitable for case reports.

We agree that audiograms alone cannot explain the etiology of SNHL in COVID-19 patients. It is precisely for this reason that our study combines clinical data and mechanistic studies involving human inner ear cells and tissues. Our new experiments involving explanted vestibular tissue from adult human inner ears demonstrate that hair cells and Schwann cells can be infected by SARS-CoV-2. In addition, we established three human induced pluripotent stem cell (hiPSC)-derived *in vitro* models of the inner ear for infection: two-dimensional (2D) otic prosensory cells (OPCs) and Schwann cell precursors (SCPs), and three-dimensional (3D) inner ear organoids. These models demonstrate that OPCs and hair cells in otic organoids are permissive to SARS-CoV-2 infection. These mechanistic studies corroborate and explain the clinical observation that otoacoustic emissions, which reflect outer hair cell function, were absent in the symptomatic ears of patients with COVID-19-induced sensorineural hearing loss. Our manuscript was amended accordingly, as explained in response to the first question of Reviewer 1.

2) Use of human iPS cell-derived cells is challenging and interesting. The problem is the use of otic prosensory cells derived from human iPS cells, not hair cells or hair cell-like cells after differentiation. The authors described that the expression of ACE2 in otic prosensory cells indicates the vulnerability of hair cells. Otic sensory progenitors can differentiate into various types of inner ear cells. It is difficult to say

the vulnerability of hair cells through the findings of otic prosensory cells. The authors should use hair cells or hair cell-like cells derived from human iPSC cells.

The revised manuscript includes new data on iPSC-derived otic organoids that include hair cells – see new Results section titled “SARS-CoV-2 infection in 3D inner ear organoids” on page 10 and new Fig. 5, new Methods section titled “Inner ear organoid differentiation” on page 21 - 22 and amended Methods section titled “Virus and infections” (page 23, last paragraph). These data demonstrating that hair cells in the inner ear organoids express ACE2 and are targets for SARS-CoV-2 are highlighted in the revised abstract and discussed in the revised Discussion section (last paragraph on page 12, entire page 13 and first paragraph on page 14).

3) An experiment of COVID-19 infection was conducted using otic progenitor cells. The use of mouse cochlear and/or vestibular explant cultures for this purpose will be more persuasive.

While using mouse cochlear and/or vestibular explant cultures is an excellent suggestion, SARS-CoV-2 cannot infect wild-type laboratory murine tissue due to poor interaction with murine analogues of viral coreceptors. A mouse-adapted SARS-CoV-2 virus has been developed (<https://www.nature.com/articles/s41586-020-2708-8>); however, we favor the use of human tissue culture models and wild type virus to best recapitulate human disease.

We were fortunate to obtain adult human inner ear tissue from indicated labyrinthectomies performed for either treatment of intractable Meniere’s disease or vestibular schwannoma resection. We then cultured these human vestibular explants (amended Methods section called “Virus and Infections”, last paragraph on page 22, continuing onto page 23) and demonstrated that hair cells and Schwann cells in explanted tissue could be infected by SARS-CoV-2 (new Results, section “**SARS-CoV-2 infection in adult human inner ear tissue**”, first paragraph on page 8 and new Fig. 2e-j). We have amended the abstract and discussion (page 13) accordingly.

Reviewer #3

Specific points

- SARS-CoV-2 infection was exclusively studied by fluorescence microscopy (staining of viral nucleoprotein or dsRNA). Especially the data in Fig. 4 would benefit from viral growth kinetics (ideally with several timepoints between 1 and 96 hpi) in the respective cell models.

The revised Results section (end of page 9 and beginning of page 8 and new Supplementary Fig. 6) now explain that “we used viral RNA in cell supernatant – detected by qRT-PCR for the SARS-CoV-2 Nucleocapsid (N) gene (<https://pubmed.ncbi.nlm.nih.gov/32396505/>) – as a surrogate to further quantify kinetics of viral replication and release. SARS-CoV-2 RNA was continuously released over the course of 72 hours, with a burst in viral production between 24 hpi and 48 hpi (Fig. 4e). Results were similar with and without normalization to 18s rRNA abundance in the supernatant (used to control for RNA extraction efficiency, noting this value may be confounded by both division and death of cultured cells). (Supplementary Fig. 6).”

New methodological details are included in the revised section called qRT-PCR (second paragraph on page 25).

- Infection rates appear relatively low (Fig. 4A and B) and especially for SCPs only a couple NP-positive cells were detected. This could be due to the low titer inoculum (MOI = 1). The authors should also test higher amounts of virus (MOI = 3 or 10 in order to “hit” more cells.

The revised results section states that “infection rates were not dramatically increased at a supraphysiologic MOI of 5, possibly reflecting activation of innate immune responses in the cells or the formation of defecting interfering viral particles at high MOI”. We further explain (in the revised second paragraph on page 9) that “we observed only rare instances of infected SCPs - rates were considerably lower in SCPs than in isogenic OPCs infected with an equivalent inoculum of virus (<1% of SCPs were infected at an MOI of 1, compared to 7% ± 1.9% of isogenic OPCs) (Fig. 4b and supplementary Fig. 5).”

Minor points

- The labeling of the panels presented in Fig. 4 does not match the figure legend and the main text. For instance, the data presented in panel C of the figure are labeled as panel C in the main text and the corresponding figure legend; panel E is not mentioned in the legend/main text.

Thank you for pointing this out. We fixed this issue.

- Please specify the statistical tests used (Fig. 3C Extended Fig. 2C and D, Extended Fig. 3A and C).

The revised manuscript includes a new methods section called “Statistical analysis” (last paragraph on p 26, continuing onto p 27), which details all statistical tests performed.

- Page 4: Please change “herpes family” and “paramyxoviridae” to “Herpesviridae” and “Paramyxoviridae”, respectively. Both terms should be italicized.

We corrected this.

- Multiple pages: Please change “nucleocapsid” to “nucleoprotein” and the corresponding abbreviation from “NC” to “NP” (main text and figures).

We corrected this.

- Multiple pages: Please change “SARS-COV-2” to “SARS-CoV-2” in the subheadings on pages 6 to 9.

We corrected this.

REVIEWERS' COMMENTS:

Reviewer #2 (Remarks to the Author):

The authors well responded to all the questions that reviewers aroused and performed a couple of important new experiments using human vestibular epithelia that were obtained from surgically removed tissues and 3D cultures of human iPS cell-derived inner ear organoids.

For critiques associated with clinical issues from Reviewer1, the authors well responded.

The revised manuscript is suitable for publication.

Reviewer #3 (Remarks to the Author):

The authors have addressed all of my points. The new data have further increased the quality of the manuscript and based on the revision made by the authors I can recommend the manuscript for publication.

Minor points:

On page 4 there is a typo in Paramyxoviridae that might have resulted from a typo in my first review.